# Relationship between Academic Stress, Physical Activity and Diet in University Students of Education

**DOI:** 10.3390/bs9060059

**Published:** 2019-06-05

**Authors:** Ramón Chacón-Cuberos, Félix Zurita-Ortega, Eva María Olmedo-Moreno, Manuel Castro-Sánchez

**Affiliations:** 1Department of Research Methods and Diagnosis in Education, University of Granada, 18071 Granada, Spain; rchacon@ugr.es (R.C.-C.); emolmedo@ugr.es (E.M.O.-M.); 2Department of Didactics of Musical, Plastic and Corporal Expression, University of Granada, 18071 Granada, Spain; felixzo@ugr.es

**Keywords:** academic stress, university students, physical activity, Mediterranean stress

## Abstract

(1) Background: Several research works have shown the relationship between physical activity (PA), adherence to the Mediterranean diet (MD) and health. Nevertheless, there are few studies that demonstrate the relationship of these habits with academic performance, and specifically with academic stress. (2) Methods: This descriptive, non-experimental, and cross-sectional study aims to analyse the associations between these variables in a sample composed of 515 university students, using as main instruments the KIDMED, PAQ-A and the Scale of Academic Stress. (3) Results: The university students analysed showed higher levels of academic stress for women, and especially in the academic obligations and communication of own ideas. In addition, it was shown that those university students that had a body mass index (BMI) associated with being overweight or under-weight were the respondents with higher levels of stress. Finally, the students with a low adherence to MD had higher scores for stress associated with the communication of their own ideas, while PA was not related to academic stress. When sex and BMI variables were controlled in the regression model, no associations were obtained between stress and diet quality. (4) Conclusions: This study shows interesting implications in the treatment of academic stress. Although stress was not associated with diet and physical activity, it was linked to a worse state of health associated with states of being overweight, being of special interest the treatment of stress in women.

## 1. Introduction

### 1.1. Theoretical Framework

The beginning of the university stage is usually accompanied by physiological and psychological changes associated with late adolescence and emerging adulthood [1]. It should also be noted that it is an important stage of life with great changes in behaviors and responsibilities derived from emancipation and the demands developed by a new university degree [2]. In fact, Arnett et al. [3] reinforce this claim by establishing that young adults suffer physiological, sociological and cultural changes, arising from the abandonment of the family home and the influences of peer groups in this period, which can be related to unhealthy habits that affect health. In this sense, healthy physical habits, such as sports practice or following a proper diet, acquire great prominence in order to avoid non-adaptive behaviors and to promote good academic performance. 

The Mediterranean diet (MD) is one of the basic elements that should be promoted at this stage [4]. This dietary model is rich in natural antioxidants and it is characterised by foods with few saturated fats. It is also based on the consumption of fruits, vegetables, fish, nuts, olive oil, vegetables and legumes. In addition, it includes a moderate intake of meats, eggs and dairy products, and a low consumption of red meats and sausages [5]. Based on the above, it should be noted that university students often have a very irregular food patterns and a low nutritional density, where the consumption of fast food and the intake of alcoholic drinks appear. Nevertheless, this does not always happen, since there are young people who are sedentary and others who are physically active, making eating and physical activity patterns totally different [6,7].

The second basic element in the promotion of a healthy lifestyle is the practice of physical activity (PA). The concept of PA is usually confused with others such as physical exercise and sport. Merino-Marban et al. [8] define PA as all movements of the body that involve an energy expenditure: walking, climbing stairs or shopping. On the other hand, whether this PA is planned and performed in a structured and repetitive way with the aim of improving physical condition, it would be physical exercise. Finally, sport refers to a PA carried out as a game or competition where different rules are followed [9,10]. In short, it could be said that both physical exercise and sports are different types of PA, which has been shown to have a multitude of health benefits at a physical level (improving body composition or reducing the risk of cardiovascular disease); at a cognitive level (improving self-esteem, decreasing stress or greater attention capacity); or at a social level (constructing social relationships or developing values) [11,12,13].

The university stage can be a stressful process due to important changes at a social level and at an educational level according to Michels et al. [14]. They also define this stress as a set of interactions between the person and situations that exceed their own resources and pose a risk to their personal well-being. Thus, academic stress will refer to any stress that occurs within the educational field, exams, academic work or oral presentations, among others [15]. In summary, it can be established that university students are characterized by living a moment of great psychological and social changes. In addition, they have very varied eating and physical activity patterns. There is a multitude of information on diet and the practice of exercise and sport. However, there is very little information about the effects of following healthy habits in academic performance and stress. 

### 1.2. State of the Question

Several research studies have shown how the practice of PA and diet can be linked to various parameters as indicators of health status at a physical level (such as body mass index or body composition); and at a cognitive level (such as perceived well-being, stress or other psychosocial factors) [16,17,18]. These variables may have a special relationship with several socio-demographic factors, such as the influence of sex. Specifically, Yahia et al. [19] have shown that men practice more PA than women in emerging adulthood, while women follow a higher quality diet. Likewise, variables indicating health status such as BMI usually present differences based on sex [20], in the same way that happens with stress [21]. Specifically, Abdel et al. [21] shows how women tend to perceive themselves as less able to face stressful events, being able to relate to healthy habits and lifestyles. For example, Hall et al. [22] state that a stressful period can generate maladaptive behavior such as poor diet or sedentary habits, which will indirectly influence the BMI.

Reviewing some interesting findings, Jones et al. [23] and Stults-Kolehmainen et al. [24] show the associations of general stress and the practice of physical activity. Specifically, Jones et al. [23] show how in a sample of adults such as those who suffered higher levels of stress were more sedentary, noting how to follow an active lifestyle could have a preventive effect. In fact, Stults-Kolehmainen et al. [24] demonstrate, through a review of more than 168 studies, how suffering from psychological stress is associated with a behavioural inhibition that predicts lower PA levels. No recent studies have been observed in the relationship between PA and academic stress specifically, but there are studies that reveal that higher levels of PA help improve academic performance [25,26]. In fact, it has been shown that the practice of physical exercise improves attention capacity, increases cerebral oxygenation and stimulates the production of endorphins, being positive in the teaching–learning processes as shown by Erickson et al. [26]. Therefore, an interesting field of study is presented given that a poor academic performance can be linked to higher levels of stress, anxiety and depression [27].

Another approach is the relationship between dietary care and states of stress especially linked to academic stress. Godos et al. [28] demonstrated how the consumption of healthy foods such as vegetables, fruit or tea helps to improve cognition and reduce negative psychological states such as depression or stress, which can be associated with academic situations. This is due to the high polyphenol content of these foods, which has an antioxidant and anti-inflammatory effect. These findings are supported by Ward et al. [29], who also reveal how this compound allows the promotion of resilience, the reduction of stressful states and negative deterioration, so it is of interest to analyse how food quality relates to academic stress suffered by university students. According with this basis, Sominsky et al. [30] talk about the bidirectional relationship between stress and eating habits, demonstrating how suffering from stress states can increase the likelihood of obesity, as well as following a healthy diet can help mitigate the negative consequences of stress states.

Considering the basis of the existing problem of academic stress in the university stage, it can be established that these can be associated with sex, different healthy habits such as PA and the quality of the diet, as well as the state of health determined by the BMI. This study presents the following research question in order to address the need to generate actions to set an active lifestyle that enables improving the negative psychological states that are developed in the university context: Is there a relationship between academic stress levels and their different dimensions with sex, BMI, diet quality, and PA practice?

Therefore, this study pursues as main goals to: a) determine the relationships between the levels of academic stress and the sex of a sample of university students of education; b) establish the associations between academic stress, BMI, diet, and PA.

Following the problems raised and the goals stated, these hypotheses are proposed:

**Hypothesis** **1** **(H1):***Women will have greater levels of academic stress*. 

**Hypothesis** **2** **(H2):**
*Those university students with a higher BMI will have a greater stress levels.*


**Hypothesis** **3** **(H3):**
*Respondents with higher scores in diet and PA will have lower levels of academic stress.*


## 2. Materials and Methods

### 2.1. Participants and Design

A non-experimental, descriptive and cross-sectional study was carried out with a single measurement in a single group. The sample consisted of a total of 515 students enrolled in the 2017–2018 course with age range between 18 and 28 years old ($\bar{x}$ = 21.58, SD = 2.72), where a 50.8 % (n = 262) were men and a 49.2% (n = 253) were women. The sample selection was done by convenience, using as selection criteria: (a) to be enrolled in some university degree of Educational Sciences during the academic year of 2017–2018; (b) be younger than 30. The exclusion criteria were: (a) not suffer any pathology that prevents the established scales being fulfilled. 

### 2.2. Measures

The main instruments used in this research were: Test of Adherence to Mediterranean Diet (KIDMED) [6]. This sale is composed by 16 dichotomous items with an affirmative or negative response; e.g.: “You eat fresh or cooked vegetables every day”, which refer to patterns related to the MD. Four of these items have negative connotations (−1), whereas the other 12 are positively valued (+1), ranging from the final score of −4 to +12. The final score is divided into three groups in order to obtain the level of adherence to MD. This scale scored an internal consistency of α = 0.812.Physical Activity Questionnaire for Adolescents (PAQ-A). This was validated by Kowalski et al. [31] and translated into Spanish by Martínez-Gómez et al. [32]. This questionnaire establishes the level of PA practiced as well as the type of PA engaged in during the last seven days. This scale allows us to obtain a summation through 10 items punctuated by a 5-point Likert scale; e.g. “In the last 7 days. What did you usually do at lunchtime (before and after eating)”. For this questionnaire, the reliability got a Cronbach’s alpha of α = 0.875.Body weight (kg) was established by means of electronic weighing scale, using the model Tanita TBF300^®^. This model needs gender, age and height variables, measuring this last one by means of a stadiometer Holtain LTD^®^ and following the protocol established by Portao et al. [33].Scale of Academic Stress. This instrument was validated by García-Ros et al. [34] in Spanish university students. For this questionnaire, the level of academic stress is assessed using 21 items which are scored with a 5-points Likert scale: e.g., “I feel stressed when I do exhibition of works in class”. Stress is grouped into four dimensions according to this instrument, which are: Academic obligations (items 1, 5, 7, 9, 10, 14 and 15), Academic marks and future expectations (items 16, 17, 18, 19, 20, 21), Interpersonal difficulties (items 8, 12 and 13) and Communication of own ideas (items 2, 3 and 4). For this instrument, a Cronbach’s alpha of α = 0.889 was obtained, this being acceptable.

### 2.3. Procedure

The collaboration of university centres was requested through an informative letter elaborated by the Area of Corporal Expression of the University of Granada. Furthermore, the written and informed consent of the respondents was requested through a document in which the nature of the study was detailed. The data was collected during regular classes in the different university campus. Different research assistants were present in the data collection in order to ensure that questionnaires were properly completed, as well as to provide guidance for the scale application and to answer different questions from respondents. Furthermore, participants did not receive any incentives. This study was approved by the Ethics Committee of the University of Granada with code “641/CEIH/2018.”

### 2.4. Data Analysis

The data analysis was undertaken using the software IBM SPSS^®^ 22.0 (IBM Corp, Armonk, NY, USA). First, frequencies and medians were used for basic descriptors, whereas the association between the variables detailed were analysed using the *t*-test and analysis of variance (ANOVA) depending on the number of categories of each variable. In addition, a regression analysis is developed in order to control the relationship between each variable. The Kolmogorov–Smirnov’s test was used in order to check the normality of data. Levene’s test was employed in order to check homoscedasticity. Finally, Cronbach’s alpha coefficient was used to analyse the reliability of the scales used, establishing the reliability index at 95.5%.

## 3. Results

Table 1 shows the levels of global academic stress, as well as the levels of the different types of academic stress according to the sex of the respondents. Women show higher levels of stress in all the dimensions measured, but two of them (Academic Marks and Future Expectations, and Interpersonal Difficulties) did not present significant differences between sexes. The greatest differences were observed in the academic obligations (3.29 ± 0.86 vs. 2.93 ± 0.95) and the communication of ideas (2.69 ± 0.94 vs. 2.26 ± 0.76), with women obtaining higher average scores. Regarding global academic stress, women presented an average of 59.62 ± 15.02 and men a score of 53.57 ± 13.82.

Table 2 indicates the levels of global stress and the different types of academic stress according to the BMI of the respondents. It is observed how significant differences were found in the dimensions “Academic Obligations” (*p* = 0.034), and “Communication of own ideas” (*p* = 0.037). Thus, it can be observed how those who had a normal BMI were those who had less stress in all the variables, and those who had a low weight or obesity were those who had greater stress in academic obligations (2.99 ± 0.86 vs. 4.03 ± 0.74) and global stress (54.82 ± 14.19 vs. 71.00 ± 11.51).

Table 3 shows the levels of global academic stress as well as the different types of academic stress according to the level of adherence to the MD of the university students. Only significant differences were found for the dimension “Communication of own ideas” (*p* = 0.045). It is observed how in this variable the stress level decreases as the adherence to the MD is improved (3.66 ± 0.46 vs. 2.32 ± 0.82). This tendency for stress is also observed in the variables “Academic marks and future expectations” and “Global stress”, although no statistically significant differences have been obtained.

Table 4 shows the global stress levels as well as the different types of academic stress according to the level of PA practice of the respondents. No significant differences were found for any of the dimensions. In the variables “Academic Obligations” and “Global Stresss” there were hardly any differences between groups, being the groups with a medium level of PA practice those that suffered more stress.

Next, the bivariate correlations between the variables under study are shown (Table 5). A positive relationship can be observed between age and BMI (*p* < 0.05, r = 0.217), as well as a negative relationship between age and stress linked to future expectations (*p* < 0.01, r = −0.306). Likewise, it was revealed that BMI was positively related to global stress (*p* < 0.05, r = 0.185) and stress associated with academic obligations (*p* < 0.05; r = 0.198), while diet was negatively related to stress linked to the communication of own ideas (*p* < 0.01, r = −0.261). Finally, note that no relationship was found with the PA and that all stress dimensions were positively related between them, showing a medium-high correlation strength.

For the regression model developed (Table 6), the R^2^ showed a value of 0.383, which explains 38.3% of the variability of academic stress that constitutes the dependent variable. In addition, a positive value is obtained, presupposing a direct relationship between the variables. It can be established that the model is valid since F = 4.068 and *p* = 0.002, so the null hypothesis is rejected and it is concluded that the linear dependence is statistically significant, so the model is adequate. Analysing the variables that constitute the model, significant differences were observed for sex (*p* = 0.003) and BMI (*p* = 0.002). In the case of sex, an inverse relationship is shown (*b* = −0.285) which determines higher stress levels in women. For the BMI a positive relationship is shown (*b* = 0.286), specifying that higher stress levels are associated with higher BMI and weight problems. Likewise, it is observed that when sex and BMI variables are controlled, no significant relationships of stress with diet and BP are observed.

## 4. Discussion

The present research aimed to analyse the associations between academic stress, sex and different healthy habits, such as the level of adherence to the MD and the level of practice of PA. Likewise, the BMI is included in order to contrast how this relates to the stress experienced by university students of educational sciences. In this way, some studies with a similar line, both in the national and international context, are those carried out by Tehrani et al. [35], Shafiei et al. [36], Vaquero et al. [37] or Beiter et al. [38]. In all of them, the relevance of considering healthy habits for the improvement of academic stress is shown.

The level of global academic stress was medium, observing the highest scores in the academic obligations, while the lowers scores were associated with personal difficulties. García-Ros et al. [34] presented similar results, although the interpersonal difficulties were slightly higher in this case. Considering the relationships between academic stress and the sex of the university students, women showed higher scores in the global factor and for the dimensions “academic obligations” and “communication of own ideas”. These findings could be justified by the premises identified by Yang et al. [39], who establish that women have a higher vocational level towards this degree, making them have a greater critical thinking and that they are more susceptible to the characteristics of the environment. In addition, they obtained the highest scores in this dimension, as indicated by recent international studies which show the superiority of womens’ performance which is associated with the level of implication and stress generated [40,41,42].

Considering the relationship between the level of academic stress and the categorization of the BMI, significant differences were observed for academic obligations and the expression and communication of their own ideas. Specifically, it could be observed that individuals with obesity had the highest score for academic stress. According to Kerr-Gaffney et al. [43] and Krafchek et al. [44], this could be due to the anxiety linked to this stressful situation, which would modify the eating patterns, which can be associated with weight status. Specifically, states of anxiety will activate the sympathetic system, accelerating metabolism and altering appetite [45]. In addition, long-term stress states can be linked to higher food intakes, which could be justified by alterations of the hypothalamus in the production of neurotransmitters to control appetite [46].

In the present study, lower levels of academic stress were observed in those respondents who followed a diet of better quality, although only statistically significant differences were shown for the dimension related to the communication of their own ideas. At this point, it is of interest to highlight two points associated with the importance of eating habits, considering therapeutic and preventive visions. First, to follow a balanced diet could help reduce levels of stress, since the body will have the essential nutrients and this will help to prevent imbalances in the production of neurotransmitters that could facilitate states of anxiety [47,48]. Second, Bektaş et al. [47] states that it may be the stress situation itself that leads to non-adaptive behaviors linked to the intake of unhealthy foods. Therefore, interest in acting according to the academic context in order to develop flexible methodologies and learning strategies that reduce stress levels should be highlighted [49]. Nevertheless, the subsequent regression model tested the relationship between all variables, controlling for the effect of sex and BMI. In this case, no significant differences were obtained between academic stress and diet, showing different results than those obtained by Bektas et al. [47]. 

The different dimensions of academic stress have not been related to the practice of PA. Nevertheless, several research studies have shown an inverse relationship between both variables. In fact, Jones et al. [23] and Holmes et al. [50] highlighted the levels of anxiety and stress decreasing when people practice physical activity. This is because the practice of PA helps decrease energy levels giving an outlet to frustration and decreasing muscle tensions [51]. It also increases the levels of endorphins—the happiness hormone—and cortisol and norepinephrine levels which are linked to stress and anxiety [52]. In addition, we have shown that exercise helps to improve depressive states and manage complex emotional processes [53].

The main limitations of this study are presented below. In the first place, it is worth mentioning the study design as a limitation, which is descriptive and cross-sectional, so that cause–effect relationships cannot be established. Likewise, it is interesting to point out the total number of respondents, since the sample has not been representative despite the fact that a large number of subjects is collected. The use of BMI as a health indicator could be highlighted, being of greater interest the use of body composition such as fat mass and lean mass. However, the most important limitation of this study is that it has not controlled some variables that can influence stress levels, such as household income or health level. As future perspectives, it would be interesting to replicate the study in a representative sample of Spanish university students. In addition, the use of a bioimpedance scale is suggested to obtain reliable BMI values and the respective values for body composition. Furthermore, using devices for the control of physical activity would be a very important addition in terms of collecting data, since it would obtain more reliable values of the activity level of the subjects.

## 5. Conclusions

In relation to the established hypotheses, it was observed that hypothesis 1 and hypothesis 2 were fulfilled, while hypothesis 3 was not satisfied. Thus, this research presents as main conclusions a medium-high level of stress in university students of educational sciences. Specifically, women show higher levels of stress linked to academic obligations and having to communicate their own ideas. Likewise, it has been observed that having high levels of stress is associated with weight problems, especially linked to obesity states. Finally, it was observed that young people who had greater adherence to MD had lower levels of stress, without finding statistically significant differences in relation to the practice of PA. Nevertheless, the subsequent regression model tested the relationship between all variables controlling for the effect of sex and BMI. In this case, no significant differences were obtained between academic stress and diet.

## Figures and Tables

**Table 1 behavsci-09-00059-t001:** Academic stress according to sex.

				Levene Test		T-Test	
	Sex	M	SD	F	Sig.	T	Sig.
**Global stress**	Women	59.62	15.02	5.451	0.711	2.335	0.021 *
	Men	53.57	13.82				
**Academic Obligations**	Women	3.29	0.86	4.687	0.451	3.125	0.032 *
	Men	2.93	0.95				
**Academic marks and future expectations**	Women	2.92	0.89	2.994	0.758	1.730	0.086
	Men	2.64	0.88				
**Interpersonal Difficulties**	Women	2.08	1.03	3.138	0.002	1.771	0.079
	Men	1.80	0.68				
**Communication of own ideas**	Women	2.69	0.94	7.705	0.065	2.766	0.006 *
	Men	2.26	0.76				

Note 1: *, *p* < 0.05; Note 2: M, Mean; SD, Standard deviation; F, F-value; T, T-value.

**Table 2 behavsci-09-00059-t002:** Relationship between academic stress and body mass index (BMI).

	B.M.I.	M	SD	F	Sig.
**Academic Obligations**	Under-weight	3.67	0.71	2.996	0.034 *
	Normal	2.99 *^b^*	0.86		
	Overweight	3.37	1.12		
	Obesity	4.03 *^b^*	0.74		
**Academic marks and future expectations**	Under-weight	3.29	0.88	1.219	0.306
	Normal	2.71	0.89		
	Overweight	2.89	0.86		
	Obesity	3.33	0.95		
**Interpersonal Difficulties**	Under-weight	2.25	0.91	0.976	0.407
	Normal	1.86	0.86		
	Overweight	2.18	0.92		
	Obesity	2.16	1.10		
**Communication of own ideas**	Under-weight	3.33	0.76	3.401	0.020*
	Normal	2.41	0.87		
	Overweight	2.43	0.83		
	Obesity	3.50	0.63		
**Global Stress**	Under-weight	67.50	12.60	2.921	0.037*
	Normal	54.82 *^b^*	14.19		
	Overweight	59.75	15.75		
	Obesity	71.00 *^b^*	11.51		

Note 1: *, *p* < 0.05; Note 2: *b*, Bonferroni test (statistically significant differences between groups); Note 3: M, Mean; SD, Standard deviation; F, F-value.

**Table 3 behavsci-09-00059-t003:** Relationship between academic stress and adherence to Mediterranean diet (MD).

	Adherence to MD	M	SD	F	Sig.
**Academic obligations**	Low	3.07	0.30	0.155	0.856
	Medium	3.15	0.92		
	High	3.05	0.95		
**Academic marks and future expectation**	Low	3.25	0.58	0.811	0.447
	Medium	2.84	0.85		
	High	2.67	0.95		
**Interpersonal difficulties**	Low	3.00	0.47	1.519	0.223
	Medium	1.90	0.85		
	High	1.95	0.91		
**Communication of own ideas**	Low	3.66 *^b^*	0.47	2.801	0.045 *
	Medium	2.54	0.90		
	High	2.32 *^b^*	0.82		
**Global stress**	Low	66.00	8.48	0.828	0.439
	Medium	57.34	14.13		
	High	54.87	15.68		

Note 1: *, *p* < 0.05; Note 2: *b*, Bonferroni Test (statistically significant differences between groups); Note 3: M, Mean; SD, Standard deviation; F, F-value.

**Table 4 behavsci-09-00059-t004:** Relationship between academic stress and physical activity.

	PA	M	SD	F	Sig.
**Academic Obligations**	Low	3.08	0.91	0.166	0.847
	Medium	3.15	0.91		
	High	3.01	1.04		
**Academic marks and future expectation**	Low	2.62	0.93	1.676	0.191
	Medium	2.92	0.80		
	High	2.66	1.06		
**Interpersonal Difficulties**	Low	2.06	0.95	0.768	0.466
	Medium	1.85	0.87		
	High	1.92	0.65		
**Communication of own ideas**	Low	2.56	0.97	0.842	0.433
	Medium	2.47	0.82		
	High	2.21	0.85		
**Global Stress**	Low	55.86	16.18	0.312	0.733
	Medium	57.48	13.25		
	High	54.50	16.50		

**Table 5 behavsci-09-00059-t005:** Bivariate correlations between the variables.

	BMI	Diet	PA	GS	S.AO	S.FE	S.ID	S.CI
**Age**	0.217 *	−0.054	0.015	−0.129	0.022	−0.306 **	−0.024	−0.163
**BMI**		0.091	−0.130	0.185 *	0.198 *	0.107	0.133	0.068
**Diet**			0.228*	−0.142	−0.078	-0.110	−0.085	−0.261 **
**PA**				−0.041	−0.052	0.055	−0.172	−0.086
**GS**					0.883 **	0.839 **	0.614 **	0.599 **
**S.AO**						0.615 **	0.400 **	0.403 **
**S.FE**							0.429 **	0.388 **
**S.ID**								0.271 **

Note 1: *, *p* < 0.05; **, *p* < 0.01; Note 2: BMI, body mass index; PA, physical activity; GS, global stress; S.AO, stress—academic obligations; S.FE, stress—future expectations; S.ID, stress—interpersonal difficulties; S.CI, stress—communication of own ideas.

**Table 6 behavsci-09-00059-t006:** Regression analysis on stress and related variables.

MODEL	Non-Standardized Coefficients		Standardized Coefficients	t	Sig.	Confidence Interval	
	B	Standard Error	Beta			Lower Limit	Upper Limit
**Constant**	53.269	14.340	-	3.715	0.000	24.871	81.666
**Sex**	−7.474	2.802	−0.255	−2.667	0.009	−13.022	−1.925
**Age**	−0.789	0.467	−0.151	−1.691	0.094	−1.714	0.135
**BMI**	1.530	0.450	0.313	3.399	0.001	0.639	2.422
**Diet**	−1.063	0.632	−0.150	−1.681	0.095	−2.315	0.189
**PA**	0.173	0.148	0.108	1.169	0.245	−0.120	0.467

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
