# Peer review of "Relationship between Academic Stress, Physical Activity and Diet in University Students of Education"

_behavsci, 2019, doi:10.3390/bs9060059_

Round 1
Reviewer 1 Report
In the manuscript, the authors reported on the relationship of physical activity, adherence to Mediterranean diet, and academic stress. A variety of data were presented, and the findings potentially have implications for physical and mental health among college students. However, there are several major problems. Please see comments below.
1. The aims of the study were not well justified. Academic stress and performance are different domains. The authors reviewed the literature of diet, physical activity, and health in Introduction, but did not establish the relationship between stress and academic performance. Moreover, the authors did not justify why to examine the effects of sex and BMI on academic stress. Given that sex and BMI affect many health indices, it is not surprising that these factors influenced the outcomes in the present study. Thus, the purpose of the study needs to justify.
2. The analyses were inadequate. First, post hoc comparisons are needed to compare significant group differences. Second, given the effects of sex and BMI on academic stress, the interaction of these variables should be considered in regression.Third, regression did not indicate important results.
3. Many statements in Discussion were speculative and not supported by the results of analyses.
Author Response
Dear the editor and reviewers,
We would like to express our gratitude for the time taken to review this manuscript and for the comments made, which we believe to be critical for producing rigorous and quality research. We have detailed below the changes made in the original article: “Relationship between academic stress, physical activity and diet in university students of education” (behavsci-492954).
Modifications have been made in the original manuscript following the reviewers’ comments. For each modification we have written: the original comment as written by the reviewer in addition to the page and line number; and the change made in response to that comment. Changes have been made using the tool “Track changes” enabling editor and reviewers to identify modifications easily.
Comment 1:
In the manuscript, the authors reported on the relationship of physical activity, adherence to Mediterranean diet, and academic stress. A variety of data were presented, and the findings potentially have implications for physical and mental health among college students. However, there are several major problems. Please see comments below.
1. The aims of the study were not well justified. Academic stress and performance are different domains. The authors reviewed the literature of diet, physical activity, and health in Introduction, but did not establish the relationship between stress and academic performance. Moreover, the authors did not justify why to examine the effects of sex and BMI on academic stress. Given that sex and BMI affect many health indices, it is not surprising that these factors influenced the outcomes in the present study. Thus, the purpose of the study needs to justify.
Response 1:
Thanks for this indication. It has proceeded to improve the introduction including:
- The link with academic performance has been eliminated, since this variable is not measured in our study.
- The relationship of stress with sex and body mass index has been explained.
- The hypothesis and objectives have been improved following the indications of academic editor and reviewer 1.
Comment 2:
2. The analyses were inadequate. First, post hoc comparisons are needed to compare significant group differences. Second, given the effects of sex and BMI on academic stress, the interaction of these variables should be considered in regression. Third, regression did not indicate important results.
Response 2:
Thank you very much for your indications. In relation to your suggestions, indicate:
- The "post-hoc" analyzes have been included in tables 2, 3 and 4 (since they are those that use ANOVA of a factor) through the Bonferroni test. The results and the discussion have been corrected accordingly.
- The regression has been done through the inclusion method of all the variables in the model obtaining acceptable adjustment indices (R2 = 0.227, explained variance: 22.7%, F = 5.49). In this way, the effect of sex and BMI is considered along with that of other variables (reason why the relationship between diet and stress disappears, since the effect of sex and BMI is being controlled). In fact, this analysis was included as suggested by reviewer 1 in a previous round. The authors believe that, although the analyzes do not report great findings, they are correct and allow to justify the objectives of the study. However, if the reviewer estimates some more appropriate analysis, please inform us so we can include it.
Comment 3:
3. Many statements in Discussion were speculative and not supported by the results of analyses
Response 3:
Thanks for this suggestion of improvement. These statements have been amended: lines 261-262, 272-274, 280, 301, 326, 332-339.
Reviewer 2 Report
In the tables, "A" is a column header. I'm assuming this is the Mean? In the ANOVA analyses, please add pairwise comparisons to test for the difference between means, instead of reporting which means show the largest difference (tables 2-4). For the non-significant results, "trend" should not be used to describe the results (page 7). Trends usually refer to marginally significant findings, not a comparison of means. That statement is not supported by the results.
Author Response
Dear the editor and reviewers,
We would like to express our gratitude for the time taken to review this manuscript and for the comments made, which we believe to be critical for producing rigorous and quality research. We have detailed below the changes made in the original article: “Relationship between academic stress, physical activity and diet in university students of education” (behavsci-492954).
Modifications have been made in the original manuscript following the reviewers’ comments. For each modification we have written: the original comment as written by the reviewer in addition to the page and line number; and the change made in response to that comment. Changes have been made using the tool “Track changes” enabling editor and reviewers to identify modifications easily.
Comment 1:
In the tables, "A" is a column header. I'm assuming this is the Mean? In the ANOVA analyses, please add pairwise comparisons to test for the difference between means, instead of reporting which means show the largest difference (tables 2-4). For the non-significant results, "trend" should not be used to describe the results (page 7). Trends usually refer to marginally significant findings, not a comparison of means. That statement is not supported by the results.
Response 1:
Thank you very much for your indications:
- Effectively. The first column of the table refers to the average. The symbol has been replaced by “M”.
- The Bonferroni test (Post-hoc) has been added to determine between which groups the differences of means exist exactly.
- - Thanks for this indication. The expressions referring to "tendencies" have been eliminated throughout the text.
Reviewer 3 Report
Dear authors,
The manuscript is well written and easy to understand but I have some doubts and recommendations for its improvement.
- I am not an English speaker but I have seen some mistakes in the text. Please revise it.
- Has the test KIDMED been validated and translated in your country? Please use specify it in the text.
Author Response
Dear the editor and reviewers,
We would like to express our gratitude for the time taken to review this manuscript and for the comments made, which we believe to be critical for producing rigorous and quality research. We have detailed below the changes made in the original article: “Relationship between academic stress, physical activity and diet in university students of education” (behavsci-492954).
Modifications have been made in the original manuscript following the reviewers’ comments. For each modification we have written: the original comment as written by the reviewer in addition to the page and line number; and the change made in response to that comment. Changes have been made using the tool “Track changes” enabling editor and reviewers to identify modifications easily.
Comment 1:
Dear authors,
The manuscript is well written and easy to understand but I have some doubts and recommendations for its improvement.
- I am not an English speaker but I have seen some mistakes in the text. Please revise it.
Response 1:
Thanks for this indication. The text has been reviewed in order to correct different typo errors.
Comment 2:
- Has the test KIDMED been validated and translated in your country? Please use specify it in the text.
Response 2:
Thanks for this indication. The KIDMED test has been validated in Spain. The original scale is cited in reference 6: Serrá-Majem, L.; Ribas, L.; Ngo, J.; Ortega, R.M.; García, A.; et al. Food, youth and the Mediterranean diet in Spain. Development of KIDMED, Mediterranean diet quality index in children and adolescents. Public Health Nutr., 2004, 7, 931-935. Doi: 1079/PHN2004556.
Round 2
Reviewer 1 Report
The revision has been improved in summarizing the literature and justifying purposes of the study. However, the main issue regarding hypotheses and data analyses remains. Unless the data are analyzed in a more proper way, the conclusions can not be supported by the results. Therefore, unfortunately, I recommend to reject the current form of the paper, but would like to read the next version if the authors are willing to reanalyze the data. See detailed comments below.
In Section 1.2, the authors reviewed studies of sex differences in physical activity, BMI, and overall health. This line of research suggests that sex modulates the relationship of variables in the current study. However, only the main effect of sex on academic stress was examined in the analyses. The interaction between sex and PA/diet should be investigated.
In the first paragraph on p.3, Stults-Kolemainen et al.'s review was mentioned. However, blood pressure level and behavioral inhibition are not necessarily related to physical activity. The findings of Stults-Kolemainen should be elaborated, in order to support the authors' statement.
The relationship between academic stress and performance was not well stated. The second paragraph on p.3 introduces diet, stress and academic performance at the beginning, but discusses the relationship between mental health and diet, which confused readers. Mental health is not directly associated with academic performance. It should either establish the relationship between stress and academic performance, or focused mental health and diet.
Hypothesis of sex difference in academic stress should be separated from Hypothesis 1. Moreover, "Those university students with greater level of stress will have a higher BMI." implicates that stress level is an independent variable and BMI is the dependent variable. In the analyses, they were the other way around. Please correct the hypotheses.
The goals of the study should be stated before hypotheses.
Did authors adjust for years in college when comparing stress levels of sexes? Because being freshmen would relate to much less stress than being seniors in college. Years in college should also be entered in the regression model.
It is not recommended to dichotomize a continuous variable. Please present correlations among stress scores, BMI, PAQ-A score, and KIDMED score.
What's correlations between BMI, PA, and diet score? If these scores are highly correlated, they should be entered into the regression model together, i.e. multicollinearity. The authors could also consider mediation analyses, which might provide more meaningful findings.
There was no measure of cognitive or academic performance in the study. Therefore, "the relevance of considering healthy habits for the improvement of academic stress and cognitive performance is shown" (p.8) is ungrounded.
On p.9 line 264, it states that "a small reduction in the level of academic stress ..." However, the study does not involve any experimental manipulations or interventions.
In limitations, the authors mentioned that they did not control for age in the analyses. Age and years in college should be available variables and entered in regression.
Author Response
EDITOR
Dear the editor and reviewers,
We would like to express our gratitude for the time taken to review this manuscript and for the comments made, which we believe to be critical for producing rigorous and quality research. We have detailed below the changes made in the original article: “Relationship between academic stress, physical activity and diet in university students of education” (behavsci-492954).
Modifications have been made in the original manuscript following the reviewers’ comments. For each modification we have written: the original comment as written by the reviewer in addition to the page and line number; and the change made in response to that comment. Changes have been made using the tool “Track changes” enabling editor and reviewers to identify modifications easily.
Comment 1:
The revision has been improved in summarizing the literature and justifying purposes of the study. However, the main issue regarding hypotheses and data analyses remains. Unless the data are analyzed in a more proper way, the conclusions can not be supported by the results. Therefore, unfortunately, I recommend to reject the current form of the paper, but would like to read the next version if the authors are willing to reanalyze the data. See detailed comments below.
In Section 1.2, the authors reviewed studies of sex differences in physical activity, BMI, and overall health. This line of research suggests that sex modulates the relationship of variables in the current study. However, only the main effect of sex on academic stress was examined in the analyses. The interaction between sex and PA/diet should be investigated.
Response 1:
Dear reviewer,
In the introduction we talk about sex, PA, BMI and health because we relate these variables to stress.
You propose that the analysis of PA, BMI and diet be included according to sex, but this is not the objective of the study. The objective of the study is to analyze the ACADEMIC STRESS that is the dependent variable, while the PA, BMI and diet act as independent variables or grouping variables in T-test and ANOVA.
We have no problem in including these simple analyzes, but as you will understand, they do not correspond to the objective of the study (to give a vison of academic stress in association with variables liked to health). In addition, the study already has 5 tables, and including 3 more would be quite confusing. Also, the reviewer 2 and 3 have approved the manuscript, believing valid the analyzes performed.
For example, we believe it is more appropriate to include age in the regression model or perform the correlations indicated in comments 6 and 7 (these analyzes have been made and we thank them for their recommendation as they improve the quality of the study).
Therefore, we hope you understand our decision.
Comment 2:
In the first paragraph on p.3, Stults-Kolemainen et al.'s review was mentioned. However, blood pressure level and behavioral inhibition are not necessarily related to physical activity. The findings of Stults-Kolemainen should be elaborated, in order to support the authors' statement.
Response 2:
Dear reviewer,
We have thoroughly reviewed the manuscript and have not found any phrase that speaks of the relationship of "blood pressure level and behavioral inhibition".
We have reviewed the reference used "Stults-Kolemainen et al" and in its results it literally states "The majority of which indicated that psychological stress predicts less PA (behavioral inhibition) and / or exercise or more sedentary behavior (76.4%). Both objective (i.e., life events) and subjective (i.e., distress) measures of stress related to reduced PA". Therefore, we believe that the premises developed in lines 84-91 are correct and they can be checked by the academic editor.
Comment 3:
The relationship between academic stress and performance was not well stated. The second paragraph on p.3 introduces diet, stress and academic performance at the beginning, but discusses the relationship between mental health and diet, which confused readers. Mental health is not directly associated with academic performance. It should either establish the relationship between stress and academic performance, or focused mental health and diet.
Response 3:
Thanks for this indication.
The lines 104-107 about mental health has been deleted. We have added another study more related to academic stress and diet.
Comment 4:
Hypothesis of sex difference in academic stress should be separated from Hypothesis 1. Moreover, "Those university students with greater level of stress will have a higher BMI." implicates that stress level is an independent variable and BMI is the dependent variable. In the analyses, they were the other way around. Please correct the hypotheses.
Response 4:
Thanks for this suggestions. Both hypothesis have been corrected (Lines 132-138 and lines 324-326).
Comment 5:
The goals of the study should be stated before hypotheses.
Response 5:
Thanks for this indication. Hypothesis have been moved after aims.
Comment 6:
Did authors adjust for years in college when comparing stress levels of sexes? Because being freshmen would relate to much less stress than being seniors in college. Years in college should also be entered in the regression model.
Response 6:
Thanks for this suggestion. “Age” has been included in the regression model (lines 274-276). However, no statistically significant differences were found.
Comment 7:
It is not recommended to dichotomize a continuous variable. Please present correlations among stress scores, BMI, PAQ-A score, and KIDMED score.
Response 7:
Thanks for this indication. The variables are categorized in order to compare the average stress scores according to the given categories.
However, we have included the correlations (Table 5) with the interval variables (total sum of PA, diet and BMI). Lines 249-271.
Comment 8:
What's correlations between BMI, PA, and diet score? If these scores are highly correlated, they should be entered into the regression model together, i.e. multicollinearity. The authors could also consider mediation analyses, which might provide more meaningful findings.
Response 8:
Thanks for this suggestion. However, BMI, PA and diet did not correlate strong with global stress (only one of its dimensions). Therefore, the indication is not carried out.
Anyway thank you very much.
Comment 9:
There was no measure of cognitive or academic performance in the study. Therefore, "the relevance of considering healthy habits for the improvement of academic stress and cognitive performance is shown" (p.8) is ungrounded.
Response 9:
Thanks for this important indication. This sentence has been corrected (lines 284-285).
Comment 10:
On p.9 line 264, it states that "a small reduction in the level of academic stress ..." However, the study does not involve any experimental manipulations or interventions.
Response 10:
Thanks for this important indication. This sentence has been corrected. Lines 307-308
Comment 11:
In limitations, the authors mentioned that they did not control for age in the analyses. Age and years in college should be available variables and entered in regression.
Response 11:
Thanks for this indication. This limitation has been deleted because age has been included in the regression as suggested the reviewer in the comment 6. We cannot include years in college because we have not these data.
Round 3
Reviewer 1 Report
The authors re-stated the hypotheses and presented additional data. Most concerns regarding the rationale and methodology of the study have been addressed. Correlations among variables also confirmed the findings that are presented in Table 2-4.
Some remaining minor issues are listed below:
Collinearity (especially Diet and PA) may still influence the results of the regression, please acknowledge in the limitations.
In conclusions, it stats "..., it was observed that hypothesis 1 and hypothesis 2 were fulfilled, while hypothesis 3 was not satisfied." Hypotheses are supported or rejected by results, rather than to be fulfilled or satisfied. Also, no need to say "it was observed". Please change the wording.
In the last sentence, it should be "..., no significant relations between academic stress and diet were obtained".